METHODS

# Topology across scales on heterogeneous cell data

**Maria Torras-Pérez**[1], **Iris H. R. Yoon**[2], **Praveen Weeratunga**[3,4], **Ling-Pei Ho**[4,5], **Helen M. Byrne**[1,6], **Ulrike Tillmann**[1,7], **Heather A. Harrington**[1,8,9,10,11]*

1 Mathematical Institute, University of Oxford, Oxford, United Kingdom, 2 Department of Mathematics and Statistics, Swarthmore College, Swarthmore, Pennsylvania, United States of America, 3 Faculty of Medicine, University of Colombo, Colombo, Sri Lanka, 4 MRC Translational Immunology Discovery Unit, MRC Weatherall Institute of Molecular Medicine, University of Oxford, Oxford, United Kingdom, 5 Respiratory Medicine Unit, Nuffield Department of Medicine, University of Oxford, Oxford, United Kingdom, 6 Ludwig Institute for Cancer Research, University of Oxford, Oxford, United Kingdom, 7 Isaac Newton Institute for Mathematical Sciences, University of Cambridge, Cambridge, United Kingdom, 8 Max Planck Institute of Molecular Cell Biology and Genetics, Dresden, Germany, 9 Centre for Systems Biology Dresden, Dresden, Germany, 10 Faculty of Mathematics, Technische Universität Dresden, Dresden, Germany, 11 Physics of Life, Technische Universität Dresden, Dresden, Germany

* heather.harrington@maths.ox.ac.uk

**Data availability statement:** All data and code are available at https://github.com/mariatorras/tda-multiplexed-data.

## Abstract

Multiplexed imaging allows multiple cell types to be simultaneously visualised in a single tissue sample, generating unprecedented amounts of spatially-resolved, biological data. In topological data analysis, persistent homology provides multiscale descriptors of "shape" suitable for the analysis of such spatial data. Here we propose a novel visualisation of persistent homology (PH) and fine-tune vectorisations thereof (exploring the effect of different weightings for persistence images, a prominent vectorisation of PH). These approaches offer new biological interpretations and promising avenues for improving the analysis of complex spatial biological data especially in multiple cell type data. To illustrate our methods, we apply them to a lung data set from fatal cases of COVID-19 and a data set from lupus murine spleen.

## Author summary

How cells are arranged within tissues is crucial to understand disease progression. Recent imaging technologies provide detailed spatial maps of tissues, creating large and complex data sets that can be challenging to interpret. Our work explores an avenue to quantify spatial data using ideas from topology, which is the mathematical field that describes shapes. Topological data analysis offers tools that capture the structure of complex data; an active area is visualising and interpreting the topological fingerprints in the original biological context. In this study, we adapt and extend topological methods to make the resulting insights more accessible. We introduce a simple visualisation that helps locate relevant features directly in the original data. By applying this method to

**Funding:** MT is funded by the Engineering and Physical Sciences Research Council (EPSRC, https://www.ukri.org/councils/epsrc/) grant EP/W524311/1, and acknowledges support provided by "la Caixa" Foundation (https://lacaixafoundation.org/en/) grant LCF/BQ/EU22/11930074. MT, HAH, HMB, UT, IY are grateful for the support provided by the UK Centre for Topological Data Analysis EPSRC grant EP/R018472/1. HAH gratefully acknowledges funding from the Royal Society (https://royalsociety.org/) RGF/EA/201074 and UF150238 and EPSRC EP/Y028872/1 and EP/Z531224/1. HMB and IY acknowledge support provided by the Mark Foundation for Cancer Research (https://themarkfoundation.org/). The funders had no role in the study design, data collection and analysis, decision to publish, or preparation of the manuscript.

**Competing interests:** The authors have declared that no competing interests exist.

tissue images from lupus murine spleen and COVID-19-infected human lungs, we show how it can highlight and quantify cell patterning that relates to disease progression. Our goal is to make these mathematical tools easier to use and understand, contributing to a growing set of interpretable methods for describing complex data.

## Introduction

The wealth of complex spatial data sets in biology requires the development of new tools. For example, high-resolution imaging techniques have greatly increased the number of biomarkers that can be simultaneously visualised in a tissue sample. Such multiplexed imaging techniques enable the identification of cell types in a tissue at a single-cell level, producing an unprecedented amount of biological data. Analysing such heterogeneous spatial data requires new tools that can handle large data sets while giving interpretable results and new insight into the biological processes at play. Current approaches for analysing biological spatial data include standard spatially-averaged metrics, such as cell counts and densities, as well as more advanced methods that make greater use of spatial information, like co-localisation metrics or neighbourhood analyses [1–5]. These methods are usually targeted at one length scale and may neglect information at other scales.

Topological data analysis (TDA) is a relatively new field in computational mathematics which uses concepts from geometry and topology to quantify the multi-scale "shape" of data. The prominent tool, persistent homology (PH), studies data across a parameter value (e.g. threshold scale). A summary of the pipeline is given in Fig 1. The input to PH can consist of the spatial locations of cells, often referred to as a point cloud. We approximate the shape of the point cloud by building a filtered simplicial complex. The output of PH is a collection of multi-scale descriptors describing clusters, loops, voids, and so on. PH is stable to small variations of the data (i.e. small changes to the data result in small changes to the topological descriptors) [6,7]. Advances in computation as well as vectorisations (which translate information from persistent homology into a form that is amenable for statistical analysis [8]) have led to an explosion of applications in computational and spatial biology [9–16]. Variations of cell-density, faulty data and resulting outliers can pose challenges for PH. To address these issues, multi-parameter persistent homology (MPH) has been introduced [17] and successfully applied in a range of contexts, including describing and quantifying immune cell distributions in cancer [18] and improving cell classification in kidney tissue [19]. However, MPH, which studies data across multiple parameters, comes at a cost. The theoretically sophisticated methodology is hard to compute and extract information from. Instead of the MPH route, here we focus on the computationally efficient 1-parameter PH to study cellular heterogeneity by introducing visualisations and alternative weightings of PH vectorisations.

First, we propose a novel visualisation tool, persistence weighted death simplices (PWDS), which enables us to visualise and interpret features in the original data sets. PWDS highlights features in a data set that may be overlooked by PH's sensitivity to outliers (e.g. regions of low density of a cell type surrounded by high-density walls of the same cell type). In particular, PWDS is able to visualise and distinguish between features in point clouds at different densities thus replicating the main advantage of MPH over PH at no additional computational cost. Next, we introduce different weightings for PH vectorisations for statistics and machine learning, with the goal of classifying data sets with more interpretability.

We showcase PWDS as a visualisation tool and vectorisation classifications in two spatial, multiplexed tissue data sets of lupus murine spleen [20] and COVID-19 human lungs [21],

which have already been processed, segmented, and annotated. Both data sets have previously been analysed in small regions using local methods to quantify cell interactions. Here, we consider either a single cell type, a group of cell types, or all cell types together. Since PH is a multiscale, local-to-global method, we offer additional tools to spatial cell data analysis. One advantage is that our approach is flexible to cell type annotations and we find this analysis is amenable to spatial distributions of cell populations in broad cell type groups. We develop a PH-based pipeline to identify cell types whose spatial structure differs significantly between healthy and diseased tissue.

## Methods

For each data set, we analyse both individual cell types and broader biologically relevant groups of cell types, treating each as a separate point cloud. We approximate the multiscale shape of each point cloud using simplicial complexes—structures composed of points, edges, triangles, and their higher-dimensional counterparts, via the alpha filtration [22,23]. For larger point clouds (i.e. those combining multiple cell types) we also use the witness filtration [24,25], which yields smaller complexes. The proportion of landmark points in witness filtrations is detailed in S1 Table. Next, we compute persistent homology (PH) for each filtration and obtain a persistence diagram, a topological fingerprint of the data. To localise and interpret the features detected by PH in the original data, we introduce a visualisation method we call persistence weighted death simplices (PWDS). Finally, we extract topological descriptors from the persistence diagram using a range of vectorisation methods, and we apply $k$-means clustering to each vectorisation to identify topological descriptors that distinguish between healthy and diseased states. Fig 1 shows the analysis pipeline for an individual point cloud. All code and data supporting this work are available at https://github.com/mariatorras/tda-multiplexed-data.

### The data sets

We analyse two data sets obtained with multiplexed imaging techniques which have already been processed, segmented, and annotated (see details in [20,21]). The input data to our pipeline is cell centroid information of a single cell type or a group of cell types. We refer to the locations of cell centroids of one or various cell types as a point cloud.

**Lupus murine spleen CODEX data.** The first data set, obtained with co-detection by indexing (CODEX), consists of multiplexed images of normal and lupus murine spleen and

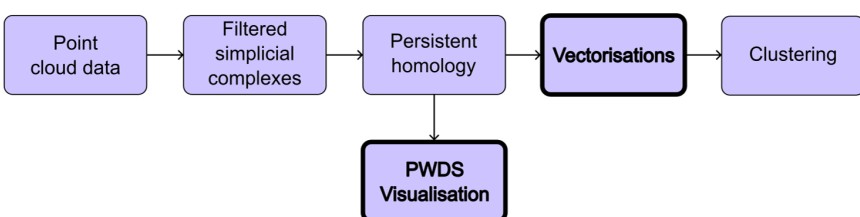

**Fig 1. Topological data analysis pipeline.** Starting from point cloud data—e.g., cell centroid locations in tissue—we construct filtered simplicial complexes to approximate the underlying geometric structure. Persistent homology is then computed to extract topological features such as connected components, loops, and voids across multiple scales. These features are vectorised, and we explore weightings that emphasise structure at different scales. The resulting representations are used for clustering. We also introduce the persistence weighted death simplices (PWDS) visualisation to enhance interpretability of the topological features.

it was taken from [20]. Nine samples were obtained from 9 mice and classified by the authors according to the state of disease progression. The authors identified 25 cell types (see Fig 2A) and used a Delaunay neighbourhood graph to quantify changes in contacts between pairs of cell types and in the environments immediately surrounding cells.

The spleen can be divided into two major anatomical regions: the red pulp, which is primarily responsible for the mechanical filtration of red blood cells, and the white pulp, which carries out an active immune response. In [20], 'the major splenic anatomic compartments were reflected in two, large, mutually exclusive clusters of positive associations between cell types, which appeared to correspond to red pulp and the white pulp'. We use these findings to classify cell types into red and white pulp (see lists on Fig 2A). Throughout the article, we use the terms 'red pulp cells' and 'white pulp cells' to refer to these broader categories. In Fig 2C we see that the white pulp roughly consists of various disconnected compartments, surrounded by the red pulp. The compartments are not closed: some red pulp cells are present in the white pulp and vice versa.

We perform a global topological analysis on the groups of cell types that form the red and white pulp, as well as on each of the 25 individual cell types.

**COVID-19 human lungs IMC data.** The second data set, obtained with imaging mass cytometry (IMC), describes the identity and location of single cells in healthy and COVID-infected human lung tissue and it was taken from [21]. The data set contains 28 lung sections from 14 COVID-19 patients (two or three sections per patient) and four lung sections from two healthy individuals. Diseased samples are categorised into three hispathology-based temporally progressive states: alveolitis (ALV), diffuse alveolar damage (DAD) or organising pneumonia (OP). The disease progresses from inflammation (ALV) to damage (DAD) to repair (OP). The images vary in size and shape. Multi-Dimensional Viewer (https://mdv.molbiol.ox.ac.uk/projects/hyperion/6567) is used to visualise the point clouds. We include a summary of the data set in S1 Fig.

Cells in the lung tissue were first classified into structural, myeloid, lymphocyte, megakaryocyte or ND (not determined), and further annotated into 49 cell types, or NA (not annotated). The authors performed a local analysis that revealed statistically significant colocations amongst immune and structural cells.

The samples infected with COVID-19 show a highly distorted lung architecture with extensive cellular infiltrate. In the samples of healthy lung tissue, the cells are organised in thin walls, of generally one cell width, that form a myriad of cavities (alveolar spaces), through which air flows into and where gaseous exchange takes place. Diseased samples are much more dense, but some regions still show some big holes formed by thick walls (visible in S1A Fig).

We perform a global topological analysis on the broad cell types of structural and immune (myeloid and lymphocyte) cells, and all cells together, as well as on each of the 49 individual cell types.

## From data to topology

We define the "shape" of data at a scale $\epsilon$ by taking the unions of balls of radius $\epsilon$ with centre at each data point. We illustrate this with an example in Fig 3A, where data points are distributed forming a loop. For small values of the scale parameter $\epsilon$, the shape we obtain is topologically equivalent to the original points, and for large values, it consists of a large blob. For intermediate values of $\epsilon$, the shape captures geometrical features of the data. In the example, the light-blue balls expand around each point, initially forming a set of disconnected balls, then merging to create a loop, which eventually fills in. Since relevant geometric features

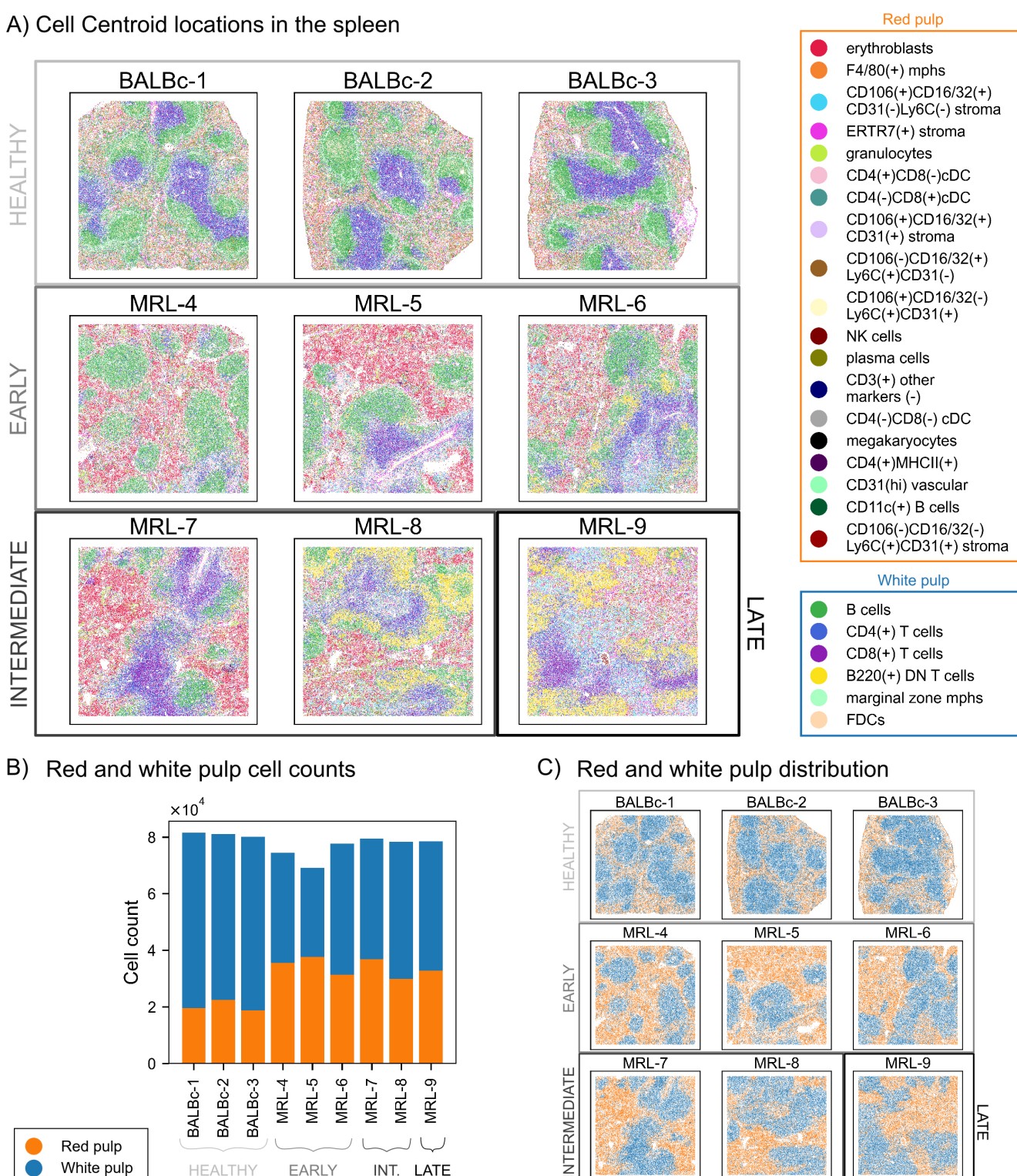

**Fig 2. Lupus murine spleen cell centroid data.** A: Spatial distribution of cell centroids for the 25 cell types in lupus murine spleen CODEX data set. The data set consists of three healthy samples (BALBc-1, BALBc-2, BALBc-3), three samples in an early stage of the disease (MRL-4, MRL-5, MRL-6), two samples in an intermediate stage of the disease (MRL-7, MRL-8) and one sample in a late stage of the disease (MRL-9). B: Cell counts of the two types of pulp for each sample (entire slide) in tens of thousands of cells. C: Point clouds corresponding to the main parts of the spleen, the red pulp and the white pulp, for each sample, classified by disease stage. The white pulp forms compartments surrounded by the red pulp.

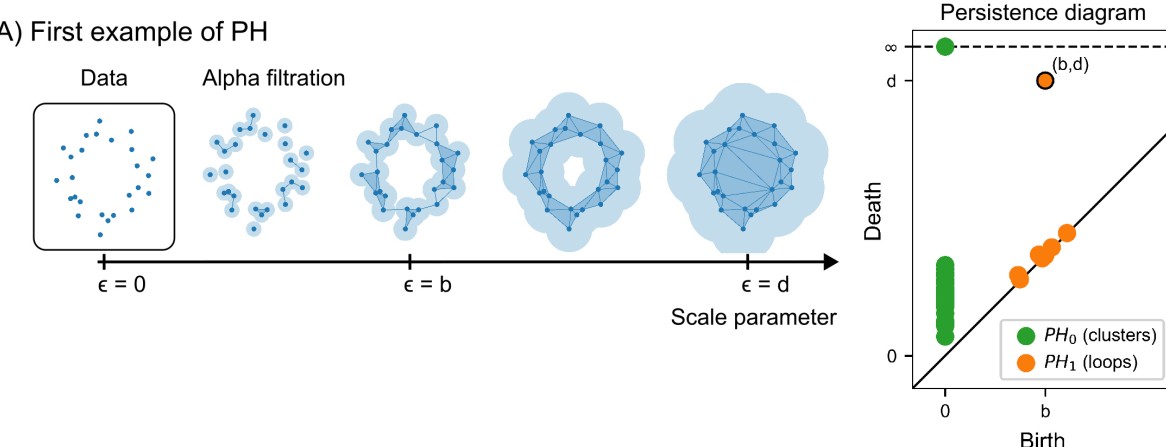

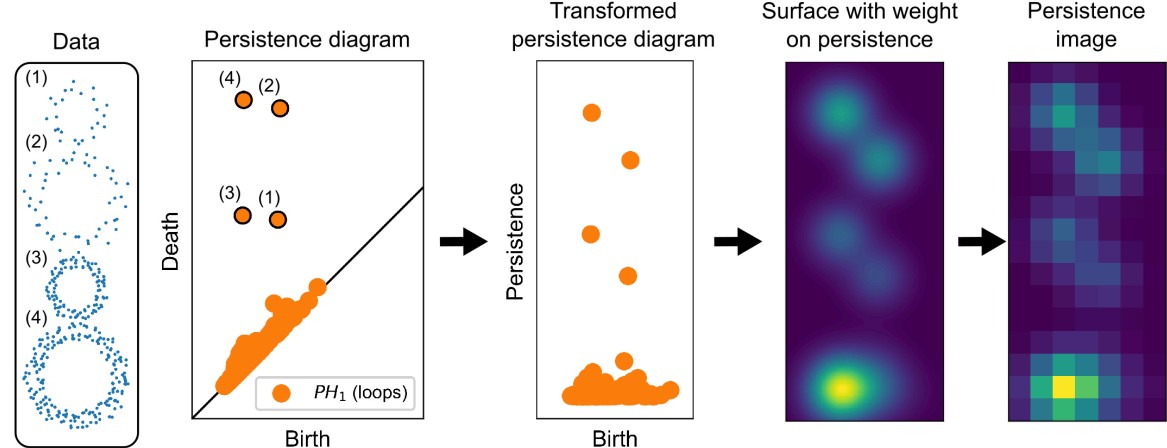

**Fig 3. Examples of persistent homology.** A: Example of the alpha filtration for a point cloud forming a loop. Corresponding persistence diagram with one persistent feature in degree 1 corresponding to the loop. B: Example of persistent homology for a point cloud consisting of four loops of various sizes and densities. For each loop, we indicate the corresponding feature in the persistence diagram. Less dense loops have larger birth values, bigger loops have larger death values. C: Steps of the computation of a persistence image with weight on persistence $w_1(b,p) = p$.

can appear at multiple scales, we avoid choosing an "optimal" value of $\epsilon$, and instead, study the family of shapes indexed by $\epsilon$.

To make the analysis computationally tractable, we replace the unions of balls by simpler combinatorial objects known as simplicial complexes, while preserving topological features. A simplicial complex $K$ is a collection of vertices (0-simplices), edges (1-simplices), triangles (2-simplices) and their generalisations in higher dimensions ($p$-simplices). Starting from the collection of data points as our simplicial complex for $\epsilon = 0$, we progressively add simplices as $\epsilon$ grows, following a proximity rule. The collection of simplicial complexes we obtain is known as a filtration (see Fig 3A). We use two constructions: the alpha filtration $A_\bullet$ [22,23] and the witness filtration $W_\bullet$ [24,25].

To build the alpha filtration $A_\bullet$, we begin by constructing the Voronoi cells of the data points. A set of data points are considered Voronoi-neighbours if their Voronoi cells intersect.

For a fixed scale parameter $\epsilon$, we construct the complex $A_\epsilon$ by first placing a ball of radius $\epsilon$ around each data point, as described above. Next, we examine the intersections of these balls for Voronoi-neighbours. If the balls of two Voronoi-neighbours intersect, we add an edge (or 1-simplex) between the corresponding points, representing their connectivity at scale $\epsilon$. To capture higher-order features beyond connectivity, we continue by adding a triangle (or 2-simplex) whenever the balls of three Voronoi-neighbours intersect. In general, we add a $p$-simplex if the balls of $p + 1$ Voronoi-neighbours have a common intersection. An example of the alpha filtration can be seen in Fig 3A.

The witness filtration $W_\bullet$ allows one to reduce the size of the simplicial complex by choosing a set of landmark points which will act as vertices, and using the rest of the point cloud to determine what simplices belong to the complex. Here, landmark points are chosen randomly. A simplex with vertices in the landmark set will be added to the simplicial complex $W_\epsilon$ if it is $\epsilon$-witnessed by some data point, in the following sense. We say a data point $w$ is a weak witness for a simplex $\sigma$ if there is a ball with centre at $w$ which contains all the vertices of $\sigma$ and no other landmark points. We relax this condition by a parameter $\epsilon$, allowing the enclosing ball to contain other landmark points, as long as they are closer to the boundary than $\epsilon$. In this case $w$ is a weak $\epsilon$-witness of $\sigma$, and $\sigma$ is included in $W_\epsilon$. We use `gudhi`'s implementation to compute the witness filtration, which uses a slightly different relaxation (see package documentation for details [26]). An example of the witness filtration can be seen in S2 Fig.

## Persistent homology

Once we have approximated the "shape" of data using simplicial complexes, we can determine their topological features—by counting connected components, loops, and voids. This is formalised through homology, a classical topological invariant. Computing homology of simplicial complexes requires only basic linear algebra, making it both accessible and efficient. In what follows, we give its definition and the intuition behind it.

To compute the homology of a simplicial complex $K$, we first build vector spaces with bases given by its 0-simplices, 1-simplices, 2-simplices, and so on. These vector spaces are denoted by $C_0$, $C_1$, $C_2$, etc. Next, we define linear maps, called boundary maps, that relate these spaces. The boundary map $\partial_2 : C_2 \to C_1$ sends each triangle (2-simplex) to a linear combination of its edges (1-simplices). Similarly, $\partial_1 : C_1 \to C_0$ sends each 1-simplex (edge) to its endpoints (0-simplices). In general, for each $n$, there is a boundary map $\partial_n$ that connects $C_n$ to $C_{n-1}$.

Intuitively, a loop or 1-hole in a simplicial complex corresponds to a collection of edges that is closed, i.e. it does not have a boundary, and it is not filled in by triangles, i.e. it is not the boundary of a collection of triangles. Mathematically, collections of edges that do not have a boundary correspond to $\ker(\partial_1)$, whereas boundaries are given by $\mathrm{im}(\partial_2)$. The homology group of degree 1 is defined as the quotient $H_1(K) = \ker(\partial_1)/\mathrm{im}(\partial_2)$, and analogously for any other degree $n$: $H_n(K) = \ker(\partial_n)/\mathrm{im}(\partial_{n+1})$. The dimensions of these vector spaces are known as Betti numbers $\beta_n$, and they quantify the number of topological features of each degree: $\beta_0$ for connected components (0-holes), $\beta_1$ for loops (1-holes), $\beta_2$ for voids (2-holes), and so on.

The approach in TDA involves analysing the shape of data across multiple scales simultaneously. To achieve this, we examine the homology of a filtration of simplicial complexes indexed by a scale parameter $K_\bullet$, like those described above. Persistent homology (PH) is an algorithm that provides a summary of the homology across the filtration, i.e, it quantifies topological features and how they persist across scales.

The inclusion of simplicial complexes in a filtration, $K_\delta \subset K_\epsilon$ for $\delta \leq \epsilon$, induces a map between the corresponding homology groups at each degree: $H_n(K_\delta) \to H_n(K_\epsilon)$, which tracks

the topological features across the filtration. A feature is born at parameter value $b$ if it first appears in $K_b$, and it dies at parameter value $d$ if it first stops being present in $K_d$. For example, the loop (1-hole) in Fig 3A appears in the filtration at $\epsilon = b$ and is filled in at $\epsilon = d$. The difference $d-b$ is called persistence, and it measures the prominence of the feature. This interpretation finds its formal algebraic foundation in the Structure Theorem of persistence modules, which states that all the homological information of the filtration can be summarised in a combinatorial way, as follows. The output of PH is a multiset of intervals $[b,d)$, each corresponding to the birth and death values of a feature. This information can be presented in a persistence diagram, where points with coordinates $(b,d)$ describe each feature in the data.

We plot the persistence diagram for the example data set in Fig 3A. We circle the most prominent feature in the diagram of degree 1, with coordinates $(b,d)$, corresponding to the large loop. Smaller loops appear between neighbouring points, as can be seen when $\epsilon = b$ in the filtration, but they have low persistence (i.e. they are rapidly filled in after they appear). If the data points represented cell centroids, the large loop would correspond to a region where cells are absent, surrounded by a wall of cells (e.g. immune cell exclusion from a tumour nest, or a structural feature like a section of a lung airway). The smaller loops would correspond to small voids formed between neighbouring cells.

PH of degree 0 corresponds to connected components. At $\epsilon = 0$, we have as many connected components as points in the data set. Each of them produces a point in the persistence diagram of degree 0 that has birth coordinate 0. As the scale parameter increases, points get connected, until a single connected component is present. With every new connection of two previously separated connected components, a 0-feature dies, corresponding to a point higher in the persistence diagram. The connected component that persists across all values is represented in the diagram as a point with death value $\infty$.

To better understand the role of birth and death values in degree 1, we consider a second data set consisting of four loops of varying sizes and densities in Fig 3B. We indicate which point in the persistence diagram of degree 1 corresponds to each loop. The denser the points forming the loop are, the earlier in the filtration the loop is formed, so loops (3) and (4) have lower birth values than loops (1) and (2). The larger the loop is, the later in the filtration the loop gets filled in, so loops (2) and (4) have higher death values than loops (1) and (3). Persistence, which is the difference between death and birth value, is larger for loop (3) than for loop (1), even though they have a similar size, and similarly when comparing (4) to (2). We say that (3) and (4) are more prominent than (1) and (2) respectively.

Finally, a key result is the Stability Theorem of persistent homology [6,7], which ensures that small variations in the data result in small variations in the persistence diagram (PH map is Lipschitz). The Stability Theorem provides the theoretical confidence of the stability of features detected by PH, making it a suitable tool for data analysis.

## Visualisation of PH: Introducing persistence weighted death simplices (PWDS)

To assess the biological significance of the topological features detected by persistent homology, we visualise their locations, especially in the case of 2D imaging data. Computing boundaries of voids detected by PH involves solving an optimisation problem which is usually computationally expensive, both in memory usage and run time. However, the approximate locations of topological features are readily available from the PH computation (e.g. `gudhi` library in Python). We exploit this information to build our visualisation without adding to the complexity of the computation.

We define persistence weighted death simplices (PWDS) as follows. We plot the simplex corresponding to the death of a feature (e.g. the last triangle that fills in a void in the filtration) on the original data. To locate specific features with similar interpretation, we choose a partition and a weighting of the persistence diagram and colour the death simplices accordingly (see Fig 4). We introduce our proposed partition and weighting, apply it to an example, and discuss well-definedness and stability of the death simplex.

We threshold features by their birth values and use colour to distinguish features formed by data points that are close (e.g. neighbouring or proximal cells), from features formed by distant data points (e.g. distal cells). Thresholds are denoted by $b_{\text{prox}}$ and $b_{\text{dist}}$. We distinguish three different regions in each persistence diagram $\mathcal{D}$:

**Red features** $\{(b,d) \in \mathcal{D} \mid b < b_{\text{prox}}\}$. Features with small birth value, that is, formed by neighbouring points in the data (e.g. proximal cells).

**Blue features** $\{(b,d) \in \mathcal{D} \mid b > b_{\text{dist}}\}$. Features with large birth value, that is, formed by distant points in the data (e.g. distal cells).

**Purple features** $\{(b,d) \in \mathcal{D} \mid b_{\text{prox}} \le b \le b_{\text{dist}}\}$. A smooth red-to-blue gradient is used to represent intermediate-scale features, ensuring stability under small variations in birth values.

We then weight each feature in the diagram by its persistence, and colour death simplices accordingly, using darker shades for more prominent features. The colouring of the persistence diagram can be seen in Fig 4A.

To illustrate, we build PWDS of a series of loops with increasing "noise" inside (see Fig 4B). These can be thought of as cell locations forming a ring structure, undergoing a process of infiltration. We consider PWDS visualisation of degree 1 PH, which measures loops in data.

We observe that most loops in cell data correspond to small voids between neighbouring cells. Based on this, we define "features formed by proximal cells" (red features) as those within the 90th percentile of birth values. We then apply a red-to-blue colour gradient to features between the 90th and 98th percentiles of birth values. The cap on the gradient avoids flattening the colour contrast due to a small number of loops with exceptionally large birth values. Features with birth values above the 98th percentile are considered "features formed by distal cells" (blue features).

To determine consistent threshold parameters across a data set, we compute the 90th and 98th percentiles of birth values for each point cloud (e.g. cell locations for a given cell type and tissue sample), and take the average across samples: $b_{\text{prox}} = \langle P_{90} \rangle$ and $b_{\text{dist}} = \langle P_{98} \rangle$. This choice proved useful for the data sets we analysed.

The PWDS of the loops can be seen in Fig 4B. Note that the size of the circumradius of a triangle is determined by its death value, while the prominence of a feature is better measured by its persistence (i.e., the intensity of the colour).

The PWDS visualisation of the first loop only contains red triangles: all features are formed by proximal points. There is one intense-red triangle (i.e. the large loop) and many small light-red triangles surrounding it (i.e. small voids formed in the thick wall of the loop).

When a few points are placed inside the loop, the persistence diagram and PWDS change drastically. New blue triangles appear, corresponding to features formed by distal points inside the loop. The red triangles are still present, but the large loop appears as a lighter red triangle: it is less prominent. Enhancing the persistence diagram with PWDS allows us to locate a noisy loop in the data that could otherwise be missed.

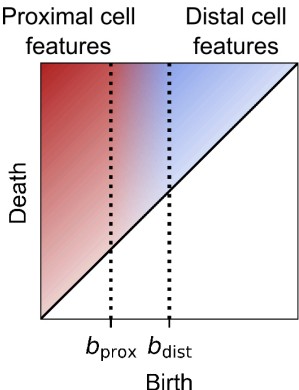

A) Colouring and weighting of the persistence diagram

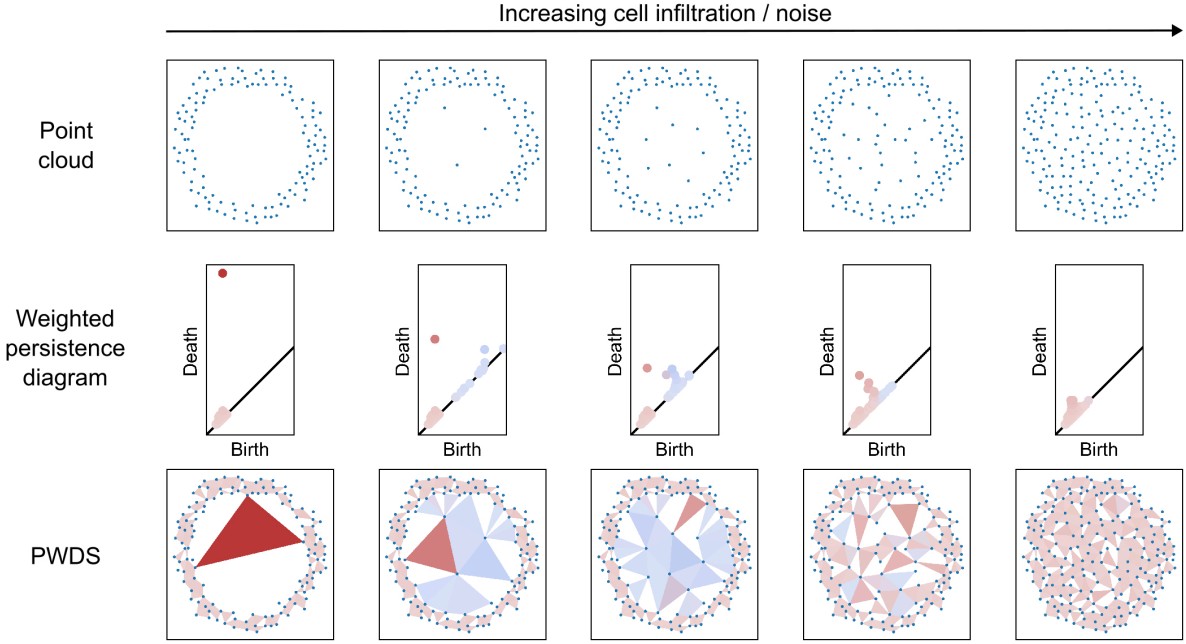

**Fig 4. Persistence weighted death simplices (PWDS) visualisation.** A: Colouring and weighting of the persistence diagram. The colouring in the diagram distinguishes features based on their birth values, with red representing features formed by proximal points, blue for those formed by distal points, and a continuous gradation of colour for features in between. The intensity of the colour reflects the feature's persistence (death minus birth), with darker shades indicating more prominent features. B: Visualisation of persistence weighted death simplices (PWDS) for five loops with increasing noise with thresholds $b_{\text{prox}} = \langle P_{90} \rangle$ and $b_{\text{dist}} = \langle P_{98} \rangle$. Each triangle indicates the approximate location of a loop detected by persistent homology with alpha complexes, coloured as described above. The PWDS visualisation of the first loop shows only red triangles, indicating features formed by proximal points, with a prominent red triangle for the large loop and smaller red triangles for voids. As noise is added inside the loop, blue triangles appear for features formed by distal points, and the large loop becomes less prominent. With increasing noise, the red triangles shrink and lighten, and eventually, only light-red triangles remain, indicating the loss of structure at the larger scale.

As we increase the level of noise, more intense blue triangles appear. Structure appears at a larger scale. At the same time, the intense-red triangle (i.e. large loop) continues to shrink and lighten. For the largest noise level, both the large loop and the separation between the two scales are lost: only light-red triangles are present.

On a more technical note, to ensure that the death simplex is uniquely determined we assume that only one simplex enters the filtration at a time. This condition holds true for the alpha and witness filtrations of point distributions in generic positions. If this condition is not met, and a feature is "killed" by two or more simplices simultaneously, `gudhi` still outputs a death simplex. However, the choice of death simplex in such cases depends on the internal, arbitrary ordering of the simplices that appear at the same filtration value.

Unlike the persistence diagram, the death simplex is not stable under small variations in the input data. To understand why, consider the case of a noisy loop, like those in Fig 4B. The death simplex for the large outer loop essentially corresponds to the triangle formed by three vertices inside the loop that has the largest circumradius, with no other points inside. If two or more empty triangles have similar circumradii, a small change in the points' positions can alter which triangle is selected as the death simplex.

Despite the instability, the death simplex will always correspond to a simplex within the interior of the topological feature, roughly indicating its position in the original data.

## Vectorisations

The space of persistence diagrams can be equipped with a metric such as the bottleneck or Wasserstein distances, but these are hard to compute, and since they heavily depend on the high persistence features, they can be too coarse for some classification tasks where differences reside in finer scales. A common approach involves mapping the space of persistence diagrams to a vector space, which is more amenable to statistical analysis and machine-learning techniques. These techniques are known as vectorisations of persistent homology. A vectorisation is called stable if its output is stable (e.g. Lipschitz) under small perturbations of the persistence diagram with respect to the bottleneck or Wasserstein distances. For an extensive survey of the vectorisations available, see [8]. In this paper, we consider the following vectorisations.

**Normalised Betti curves (one vectorisation).**  Given a filtration of simplicial complexes $K_\bullet$, the Betti-$n$ curve is a function that assigns to each parameter value $t$ the Betti-$n$ number of $K_t$ —the number of topological features of degree $n$ present in $K_t$ (e.g. the Betti-1 curve represents the evolution of the number of loops in the filtration). It can easily be computed from the persistence diagram.

To account for differences in the sizes of point clouds, we normalise the curve by the total number of features in the persistence diagram. We obtain a normalised Betti curve, which represents the proportion of features that are present at a given parameter value. Note that normalisation here does not mean that the area under the curves is equal to 1. The area under the curve corresponds to the mean persistence of features.

**Elementary statistics (three vectorisations).**  A direct approach to analysing a set of persistence diagrams is to consider elementary summary statistics from the diagrams. We obtain three vectorisations using this approach. We consider the distributions of birth, death, and persistence values. To capture the shape of these distributions, we compute the 10th, 25th, 50th, 75th and 90th percentiles for each, and take their values as the entries of a vector.

**Persistence images: exploring the role of weightings (five vectorisations).**  Persistence images are a stable vectorisation of persistent diagrams that were introduced in [27]. Intuitively, persistence images are a way of "smoothing out" persistence diagrams. First, we change the persistence diagram from birth-death coordinates $(b,d)$ to birth-persistence coordinates $(b, d-b)$. We then place a Gaussian of a certain variance centred at each point of the persistence diagram. We obtain a surface by summing all the Gaussians, weighted by a function that depends on the point where they are centred. To ensure stability, this weighting

function must be non-negative, zero along the horizontal axis, continuous and piecewise differentiable. Finally, we fix a grid of a certain resolution, and we integrate the surface over each box. The coordinates of our vector will be the values of these integrals. The steps of this pipeline are exemplified in Fig 3C.

Persistent images have been experimentally shown to be robust to the choices of variance of the Gaussian and of resolution of the grid (see section 6.2 of [27]). Intuitively, the variance and resolution parameters measure how much of the information contained in the persistence diagram is "blurred". In extreme cases, a single square grid or a very high variance would only record the number of features, and a very small grid with very small variance would be basically equivalent to the original diagram. There is an intermediate range of parameter values for which persistence images capture the coarse features of the persistence diagram, and can be suitable for classification tasks. Our choices of parameter values can be found in S2 Table. We consider five different weighting functions which aim to detect differences at various scales. In what follows, $b$ will correspond to the birth of the feature, and $p = d - b$ to its persistence.

1. **Flat function (one vectorisation).** This function weights all features in the diagram almost equally, while minimally satisfying the stability condition.

    We look for a function that is zero along the horizontal axis, increases and plateaus quickly. A simple and natural choice is a linear ramp that plateaus at $p_0$, that is:

$$
w_{\text{flat}}(b,p) = \begin{cases} p/p_0 & \text{if } 0 \leq p \leq p_0, \\ 1 & \text{if } p > p_0. \end{cases}
$$

    The parameter $p_0$ determines the minimum persistence for features in the data to be considered relevant. Below this threshold, features are considered noise. The choice of $p_0$ needs to be adapted to the scale of the data set, so any choice needs either prior knowledge of this scale, or needs to depend on the specific data set. We argue that there is generally a "natural" small scale $s$ in a point cloud coming from real data. For biological tissues, for example, there exists a typical distance between cell centroids in a given type of tissue. We propose to set $p_0$ to be the persistence of a "minimal feature" at this scale $s$.

    We approximate $s$ in the data by the average nearest neighbour distance across samples, when considering all cell types together. For degree 0, we take $p_0 = s/2$, the typical distance at which neighbouring cells are connected. For degree 1, we consider a loop formed by three cells forming an equilateral triangle of side $s$, which has persistence $p_0 = (1/\sqrt{3} - 1/2)s$.

    In some cases we will compute the persistent homology of subsets of the original point clouds, for example, isolating a single cell type, which changes the distances between neighbouring cells. We will nevertheless fix the parameters $s$ and $p_0$ as computed for the whole point clouds.

2. **Functions that weight the persistence value (two vectorisations).** We consider the functions $w_1(b,p) = p$ and $w_2(b,p) = p^2$, which weight higher persistence features, in order to explore the large scale structure.

3. **Functions that weight persistence and birth values (two vectorisations).** We consider $w_3(b,p) = bp$ and $w_4(b,p) = bp^2$, which weight higher persistence features and features with a higher birth value. Heuristically, the goal of these functions is to detect noisy loops.

When a few points are placed inside a point cloud that originally formed a loop, the persistence diagram for the alpha filtration changes drastically, with the original feature now having a much lower death value, and some new low persistence features appearing at higher birth values (see the first two columns of Fig 4B). This non-robustness of persistent homology is usually regarded as a downside, but here we propose a vectorisation that aims to highlight this large-scale noise in order to differentiate noisy loops from smaller clean loops.

We compute all 9 vectorisations for every persistence diagram of degree 1. For degree 0, since all features appear at parameter value 0, death values coincide with persistence and we only need to consider the percentiles of persistence values and the first two types of weightings for the persistence images.

### Clustering

We use $k$-means clustering to identify those topological descriptors that differentiate between healthy and diseased samples in the two data sets, as well as classify the stages of the disease.

We evaluate the clusterings at two levels. If they match the biological classification, they are considered correct, and we use the silhouette score to compare their strength. If the clustering is not perfect, we may also measure its similarity to the biological classification using the Rand score.

### Results

Our analysis reveals the inherent heterogeneity of spatial biology data, with cell patterns strongly associated with transitions from healthy to diseased tissue states. We utilise machine learning vectorisations of shape descriptors and introduce new visualisations to enhance our understanding of these transitions. Specifically, we compute persistence diagrams, multiscale descriptors that capture the "shape" of data, and introduce a novel visualisation, Persistence Weighted Death Simplices (PWDS).

In our first data set —images of normal and lupus murine spleen— PWDS highlights infiltration differences of red and white pulp cells between healthy and diseased samples (Fig 5). Further analysis with normalised Betti curves reveals differences in small-scale cellular arrangements in the pulp and some individual cell types, notably B cells (Fig 6). An exhaustive analysis of all individual cell types with a range of vectorisation techniques shows changes in the topology across various scales for ten out of 25 cell types in the spleen (S3 Table).

In our second data set —images of healthy and COVID-infected human lungs— PWDS visualisations capture the changes in the coarse structure of the tissue in disease, and normalised Betti curves quantify the changes in the small-scale patterns due to infiltration (Fig 7). We observe that clustering based on the topology of Endothelial cells closely aligns with clinical classification of disease stages (Fig 8).

### PWDS visualisation reveals red pulp cells infiltrate white pulp in lupus

We study the spatial patterns of two cell populations, the broad groups of cell types corresponding to red pulp and white pulp, in mouse spleen. A natural question is whether the cell patterning, quantified by topological data analysis, differs between wild-type and lupus (diseased) mice. We first visualise the topological fingerprint from persistent homology with the proposed visualisation, persistence weighted death simplices (PWDS). Rather than only analysing the topological fingerprint, we interpret these topological features in the original

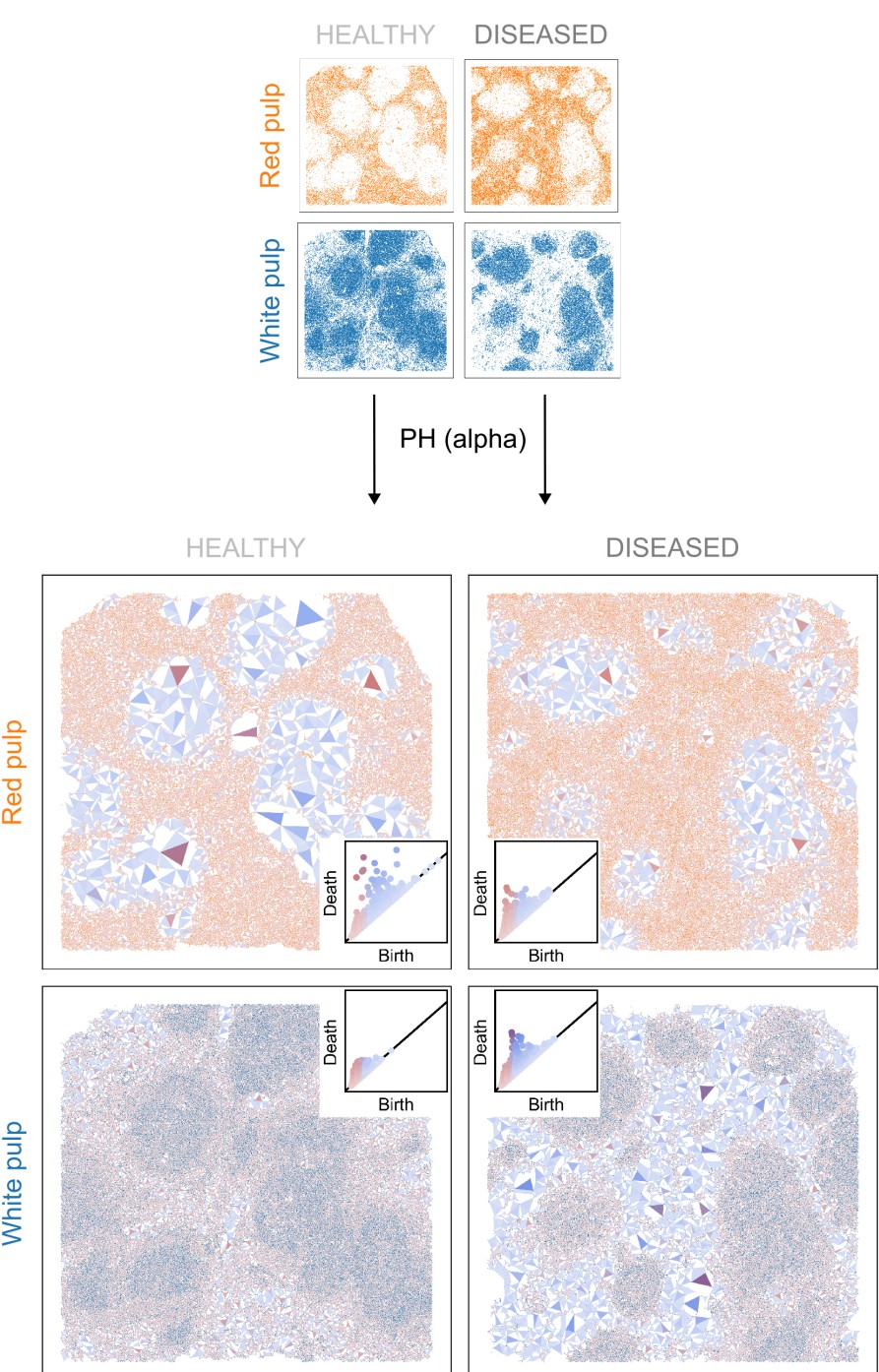

**Fig 5. PWDS of red and white pulp cell populations in the murine spleen tissue.** Visualisation of PWDS for the red and white pulp in a healthy sample (BALBc-1) and a diseased sample (MRL-4). Triangles correspond to loops detected by PH with alpha complexes. Red triangles correspond to loops formed by proximal cells, blue ones to loops formed by distal cells, and the intensity of colour is set according to the prominence of the feature. The parameters of the colour gradation depend on the cell type as in the example in the Methods section: $b_{\mathrm{prox}} = \langle P_{90} \rangle$ and $b_{\mathrm{dist}} = \langle P_{98} \rangle$. PH detects the coarse structure of the spleen, i.e. the noisy holes left in the red pulp by the white pulp as prominent loops (intense-red triangles). The presence of more numerous small light-blue triangles in the red pulp PWDS of diseased samples indicates a higher level of infiltration of the red pulp cells inside the holes left by the white pulp.

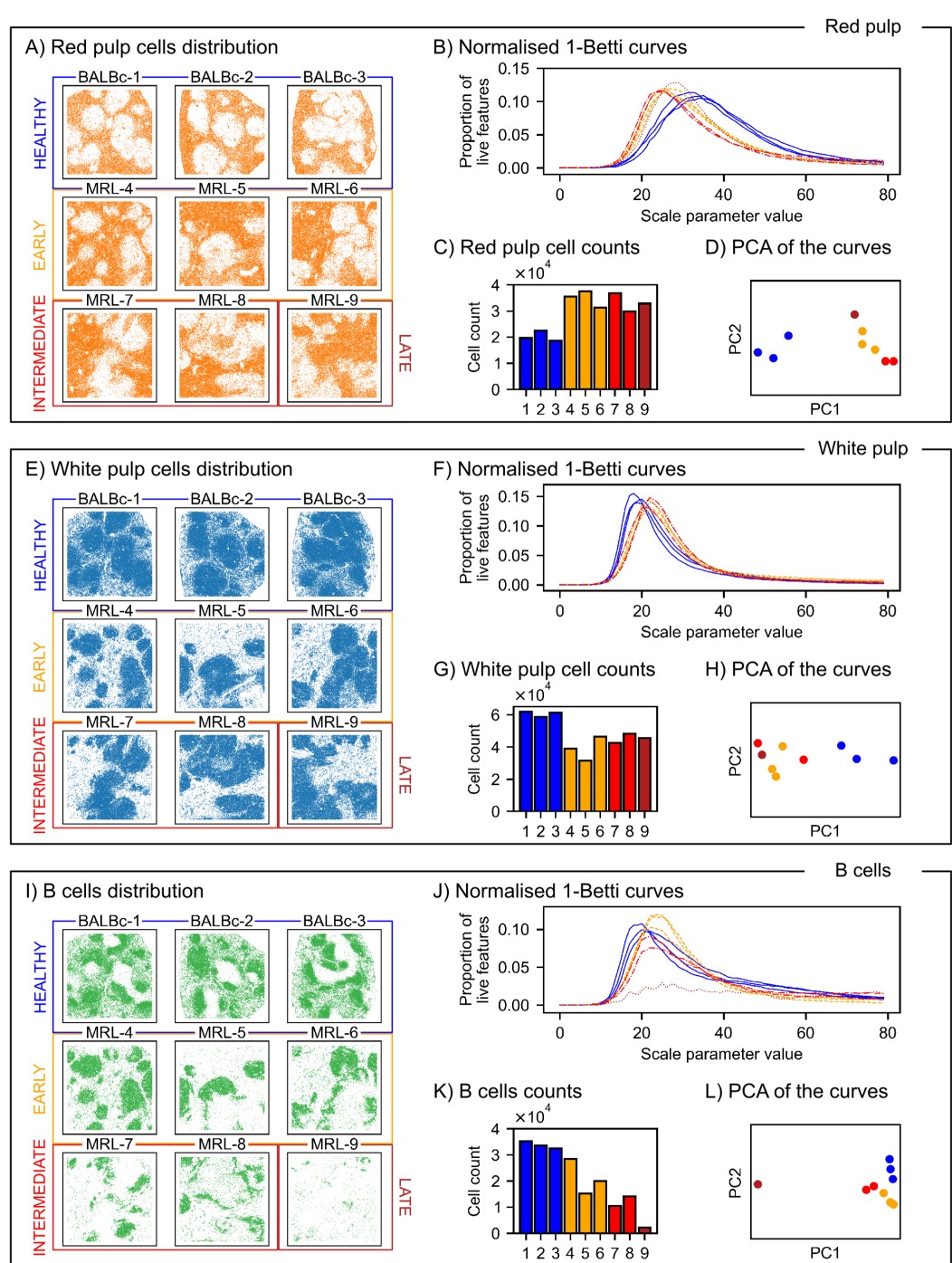

**Fig 6. Betti curves of red pulp, white pulp, and B cells populations in the murine spleen tissue.** A: Red pulp cell locations in the lupus murine spleen CODEX data. B: Red pulp normalised Betti-1 curves, which count the proportion of loops that are alive at a certain parameter value, relative to the total number of loops detected by PH (alpha). The peaks of the curves are due to a large proportion of cells within dense regions where many small-scale loops form. The location of the peak measures the typical scale of these dense environments. Differences in the location of the peak indicate an increase of density within the dense regions of the red pulp in disease. C: Cell counts of the red pulp cells for each sample. D: PCA plot of the red pulp normalised Betti-1 curves of the alpha filtration, where we observe two separated groups corresponding to healthy and diseased samples. E, F, G, H: same plots for white pulp cells, where the opposite phenomena is detected by the Betti-1 curves: a decrease in density in the dense regions of the white pulp in disease. I, J, K, L: same plots for B cells, where apart from a displacement in the peaks due to a decrease in density within the B follicles, we further identify a decrease in the height of the peak of Betti-1 curves as the disease progresses, signalling a reduction in the size of the B follicles.

## A) Examples of PWDS visualisation for all cells in the lungs

## B) Normalised 1-Betti curves for all cells

**Fig 7. PWDS and Betti curves for all cell types together in the human lung tissue.** A: Visualisation of PWDS for all cell types together with a choice of six representative samples out of 32. The visualisation corresponds to loops in PH alpha, with threshold parameters $b_{\text{prox}} = \langle P_{90} \rangle$ and $b_{\text{dist}} = \langle P_{98} \rangle$. We observe the extensive cellular infiltrate in the lung tissue infected with COVID-19, indicated by small light-red triangles. In healthy samples PH detects the cavities in the lungs formed by thin walls of cells, indicated by intense red-triangles. For some COVID-19 samples, holes within the cellular infiltrate are also detected. Since there is not much noise inside the holes in this data set, there is a small number of blue triangles. Most of them correspond to either holes of concave shapes, or to holes whose walls are not completely contained in the sample, so they are formed by distal cells. B: All cells normalised Betti curves of degree 1 of the alpha filtration. We observe that diseased samples have a very similar behaviour to what we observed in the spleen data set, a peak in the small-scale due to the presence of large dense regions. The Betti-1 curves for the healthy samples do not have a pronounced peak at small scales, since the thin walls do not contain a large number of small-scale loops, but they have heavier tails due to the large proportion of large-scale holes.

## A) Clusters based on Endothelial cells topology

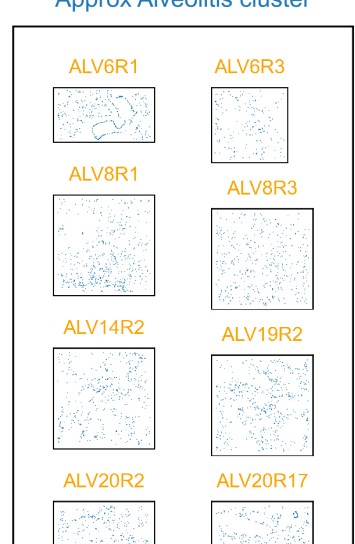

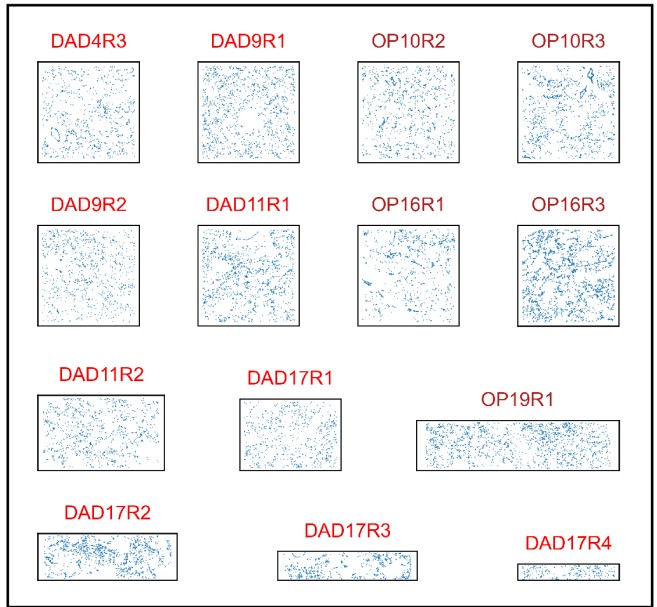

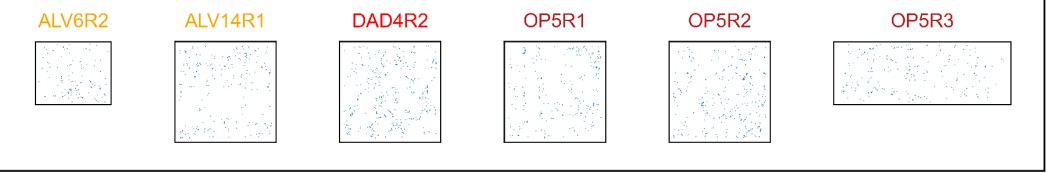

## B) PCA of the topological descriptor with a clustering of higher Rand Score

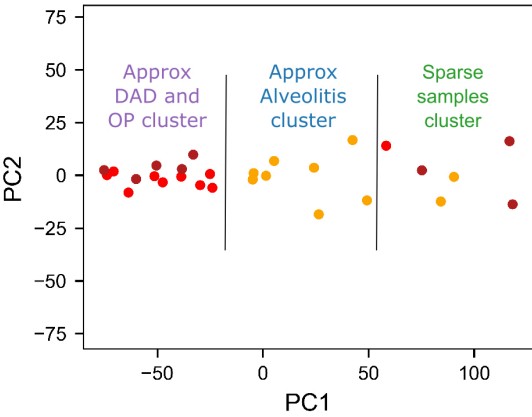

## C) Relation of the clustering to the density of Endothelial cells

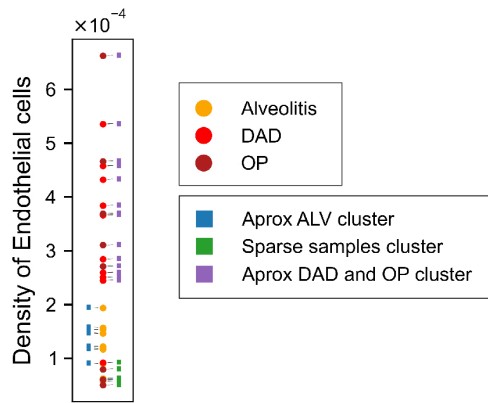

**Fig 8. Clustering based on the topology of Endothelial cells in the human lung tissue.** A: Clustering based on the topological descriptor that attains the best Rand score when compared to the biological classification: percentiles of death values of degree 1 PH (alpha) of Endothelial cells. We observe that the Alveolitis samples are approximately separated from the other two stages, and the third cluster contains sparser samples. B: PCA of the topological descriptor from Endothelial cells that attains the best Rand score, with the clusters separated by black lines. C: For each sample, we plot the density of Endothelial cells. We indicate by color yellow, orange or red the stage of the disease, and by a label of color blue, green or purple the label in the clustering we obtain. The clustering corresponds approximately with thresholding by density.

data. The location of voids surrounded by a loop of cells are visualised as triangles. Red triangles represent voids surrounded by loops of proximal cells, blue triangles represent those formed by distal cells, and intermediate voids are shown with a red-to-blue gradient. Colour intensity increases with prominence of the feature. The thresholds for classifying features as formed by proximal or distal cells are computed for each cell type, based on the average 90th and 98th percentiles of birth values in the persistence diagrams (as in the example Fig 4B).

The PWDS visualisation for red pulp cells in Fig 5 allows us to identify the level of heterogeneity of red pulp cells inside the white pulp in healthy and disease tissue. The overall architecture of healthy and diseased samples is similar; they both have dense regions of red pulp cells and voids (i.e. where there is white pulp). The dense regions of red pulp cells can be visualised in the PWDS as small light-red triangles, corresponding to small loops formed by proximal cells inside the red pulp. The voids or absence of red pulp cells (i.e. where there is white pulp) are shown as intense-red triangles. Since there are some red pulp cells inside the white pulp in all samples, some features are formed by distal cells, which we observe as many blue triangles. The infiltration of red pulp cells in the white pulp voids differs between health and disease. In the healthy sample, the colour of the red triangles inside the holes is more intense, indicating larger, more prominent features than the diseased sample. The diseased samples have more light blue triangles, indicating more red pulp cells in white pulp tissue (i.e. infiltrating the red pulp voids). A synthetic example of "healthy" behaviour can be seen in the second column of Fig 4B whereas behaviour in diseased samples resembles the third and fourth columns of Fig 4B, where we progressively increase cell infiltration.

We detect the opposite phenomenon for the white pulp cells, whose presence in the red pulp decreases in the diseased samples (Fig 5). In this case, the PWDS visualisations of both healthy and diseased samples contain less prominent features formed by proximal cells (i.e. points at the top left of the persistence diagrams). These findings are consistent with the architecture of the spleen, where the red pulp surrounds the white pulp. When the white pulp is removed from the red pulp, prominent voids appear, but the opposite is not true. Outside the white pulp compartments, we only observe blue triangles, corresponding to features formed by distal cells. These are smaller and lighter in the healthy samples, indicating a greater presence of white pulp cells in the red pulp.

## Betti curves distinguish proximal cell patterns in lupus

We study the cell patterns formed by proximal cells in dense regions of red and white pulp. Instead of selecting a threshold distance to analyse the small features, we sweep through the scale parameter (e.g. the range of distance between proximal cells). We then record the number of features (e.g. 1-dimensional voids) at each parameter value and then normalise it by the total number of features. We compute this multiscale descriptor, a normalised Betti curve, for red pulp cells in Fig 6B and white pulp cells in Fig 6F.

In both cell populations, the majority of loops are formed by proximal cells in the densest regions, corresponding to light-red small triangles in the PWDS visualisations. Therefore, the normalised Betti curves have their strongest signal on the scale for which these loops are present. The value of the scale at which a Betti curve peaks provides a measure of the size of the small voids formed by proximal cells in dense regions of each type of pulp.

For the red pulp cells, the peak of the Betti curves occurs at a larger scale paramater value in the healthy samples (i.e. the peaks of the diseased samples are further left). We interpret the different peak values to indicate that the proximal red pulp cells form larger cavities in the healthy samples.

Since the analysis is performed on each cell population, more white pulp cells in the dense red pulp regions means that there are many holes in the pattern of red pulp cells. The findings from the normalised Betti curves are thus consistent with the PWDS visualisation in Fig 5A.

For the white pulp, we observe the opposite behaviour: proximal white pulp cells form larger cavities in the diseased samples, which is consistent with earlier observations that infiltration of the red pulp into the white pulp is greater in disease than health.

We compute Betti curves for the 25 individual cell types in the spleen, and observe a similar behaviour for the densest populations, that is, the Betti curves have a prominent peak at a small scale, located at different values for healthy and diseased samples. This is the case for F4/80(+) macrophages, granulocytes, and B cells; the later will be further discussed in the next section. For erythroblasts and CD4(+) T cells, the same phenomenon is observed, but one of the diseased samples is sparser than the rest so its peak occurs at a larger value. The case of CD4(+) T cells (S3B Fig) is of particular interest because in this case the Welch test does not identify a statistically significant difference in cell density between health and disease (p-value ≈ 0.16), showing how PH can detect spatial differences in samples of similar density.

For B220(+) DN T cells (S3F Fig) and CD106(+)CD16/32(+)CD31(-)Ly6C(-) stroma, the dense regions expand with the disease progression, which causes the peak to move from a larger to a smaller parameter value. For some early diseased samples we observe two peaks in the Betti curves, one at the larger scale of the sparser regions, and one at the small scale of the densest regions. For other dense cell types such as ERTR7(+) stroma or CD8(+) T cells the Betti curves do not differ significantly between health and disease.

For Betti curves with peaks located at a similar value, the height of the peak indicates the size of the dense regions relative to the total area in which those cells are found. A larger proportion of cells in dense regions means a larger proportion of small voids and, hence, higher peaks.

We can observe this effect on the Betti curves of B cells (Fig 6J). In the healthy samples, B cells densely populate specific compartments of the white pulp known as B follicles. We note that the peak of the diseased curves occurs at a larger scale parameter value than in the healthy samples, indicating that B follicles are less densely populated in diseased samples. We also observe a progressive decrease in the height of the peaks from the early to the intermediate and then late stage. This change corresponds to a reduction in the size of the B follicles, finally ending in a very small population of B cells in the late diseased sample, where the small loops formed by proximal cells no longer dominate the behaviour of the Betti curve, which actually has a much higher peak at a larger scale.

## Topological vectorisations of ten cell types distinguish lupus samples

We next analyse all 25 cell types and two broader groups of cell types with the topological data analysis pipeline introduced in the Methods section. The pipeline takes in a sample, applies a filtration to each cell type, computes persistent homology ($PH_0$, clusters, or $PH_1$, voids), and then computes a vectorisation: Betti curves, three persistence statistics, and persistence images with five choices of weights that give importance to different types of multiscale features. From each sample we obtain a list of 383 topological descriptors that capture geometric information of cell populations at different scales.

We investigate which topological descriptors distinguish the healthy and diseased samples based on the cells' spatial structure. We apply the $k$-means clustering algorithm with two and four clusters to each of the topological descriptors, and record those that yield a correct clustering. A clustering is considered correct if the resulting clusters align with biologically meaningful groupings. Specifically, a 2-clustering is considered correct if it perfectly separates

healthy and diseased samples, and a 4-clustering is considered correct if it perfectly matches the classification into disease stages: healthy, early, intermediate, and late. We then use the silhouette score—where a high score indicates high-quality clustering—to assess the significance of these findings (see S3 Table). We find that ten cell types, together with red and white pulp cells, form significantly different spatial patterns between health and disease. Two cell types show significantly different spatial patterning between disease stages: B cells and CD11c(+) B cells (a sparse cell type).

As explained in the previous section, Betti curves of degree 1 capture differences in the size of small-scale patterns in cell populations that form dense regions. Betti-1 curves yield a correct 2-clustering for red and white pulp, as well as for F4/80(+) macrophages, granulocytes and B220(+) DN T cells. For B cells, the degree 1 Betti curves yield a correct 4-clustering into disease stages. As discussed above, the Betti-1 curves detect how the size and density of the B follicles decreases as the disease progresses (Fig 6J).

The Betti-0 curves measure how many clusters of cells exist at each scale parameter value, relative to the total number of cells. The Betti-0 curves also achieve a separation in the cases we mentioned, except for white pulp cells and B cells. Betti-0 curves outperform Betti-1 curves at detecting the formation of clusters of megakaryocytes in the diseased samples, with silhouette scores of 0.73 and 0.42 respectively (S3J Fig).

We also consider persistence images, which consist of a Gaussian smoothing of the persistence diagram, weighted to highlight different scales. Flat weighted images give correct clusterings for approximately the same cell types as the Betti curves of their respective degree. This is to be expected since for dense cell types both metrics are dominated by small-scale features.

We also consider weighting functions that depend on persistence and birth to quantify differences at large-scale, like those observed in the red and white pulp via the PWDS visualisation. We obtain correct 2-clusterings for most of these weights in degree 1 for red and white pulp. They also give a correct classification for other dense cell types which preserve some of the large-scale structure of the red pulp, such as F4/80(+) macrophages and granulocytes, and some sparser cell types like CD4(+)CD8(-)cDC. Marginal zone macrophages, which densely concentrate around the compartments of the white pulp, achieve a weak correct clustering too.

The percentiles of persistence values of degree 0 PH, which can be regarded as statistical summaries of the Betti-0 curves, give a correct clustering for the same cell types as Betti-0 curves (except for a very sparse cell type). A similar result occurs in degree 1 with the other elementary statistics. Elementary statistics (percentiles of birth, death and persistence values) generally achieve better quality scores due to their reduced dimensionality.

Among the statistical descriptors we find the second correct 4-clustering, obtained from the distribution of CD11c(+) B cells. This cell type is very sparse (between 33 and 260 cells per sample, representing less than 0.2% of the total number of cells). Upon further investigation of the distributions of degree 1 persistence values, we find that this result is an artifact of the sensitivity to outliers of higher percentiles (90th) for small populations (see S5 Fig for details).

## Betti curves quantify the cellular infiltrate in COVID-19 lungs

The extensive cellular infiltrate in COVID-19 lungs is captured by persistent homology. Our proposed visualisation of persistence, PWDS, highlights the structural voids in healthy lung samples (Fig 7A). These cavities are quantified by persistence and visualised as large intense-red triangles. We note that some of the diseased tissue (e.g. COVID sample 19 ROI 2) retains large cavities surrounded by dense regions of cells, visualised by large red triangles.

The distinction between these diseased samples and the healthy ones resides at the small scale. Small voids formed by proximal cells, visualised by small light-red triangles, feature prominently in diseased samples with extensive cellular infiltrate. Conversely, the healthy tissue is formed by thin walls and no small loops are present.

The small scale shape differences can be identified by the profile of the Betti-1 curves (see Fig 7B). The Betti-1 curves have a similar shape to those in the spleen data set (see Figs 6 and S3). Here, the cells in diseased samples have a large relative difference between the number of small and large features: the Betti-1 curves of diseased cell distributions peak at a small parameter value and then rapidly decrease at larger parameter values. Conversely, the Betti-1 curves of the healthy samples have a smaller peak and plateau at a higher value of the scale parameter: they have a larger proportion of medium-sized loops.

### Topology of Endothelial and immune cells separates diseased lung samples

We performed an exhaustive analysis of weighted vectorisations of shape descriptors for all cell types (i.e. 548 descriptors per sample, from 50 cell types and three groups of cell types). The descriptors that best separate healthy and disease samples (see S4 Table) align with a recent methodological study of topological vectorisations [8].

We found that descriptors that achieved separation between the groups came from either the all cell types point clouds or only the immune cells point clouds. To rule out a possible confounding effect due to gross structural differences between health and disease samples, we performed a finer clustering of the 28 diseased samples with $k = 3$-means clustering. We identified the topological descriptors that best separated the samples by computing the Rand index, which measures the accuracy of the clustering compared to the biological classification.

The highest Rand indices were achieved by topological descriptors derived from Endothelial cells (see S5 Table). The clustering with the highest score separates most samples in the Alveolitis stage from those in the DAD and OP stages, with a third smaller cluster containing a few samples where endothelial cells were specially sparse (Fig 8A and 8B). Comparing the clustering with the density of endothelial cells in each sample (Fig 8C), we see that the clusters correlate with density. However, clustering by density does not produce the same classification.

### Discussion

Topological data analysis is ideally suited for analysing complex spatial biology images due to its interpretability, stability, flexibility and integration with ML vectorisations [8,27]. To complement the identification and classification of topological features in a persistence diagram, we propose an interpretable visualisation of these features in the original data: persistence weighted death simplices (PWDS). In our study, we distinguish loops formed by proximal and distal cells in the original tissue data and weight them by prominence. This allows us to locate prominent voids in tissue and their varying degrees of infiltration, which relate to disease stage.

Betti curves have proven to be highly effective and interpretable for describing connectivity of data point clusters (Betti-0s) and 1-dimensional voids in data (Betti-1s) [28]. We demonstrate that normalised Betti-1 curves effectively detect density differences in non-homogeneous tissue samples, due to their sensitivity to small-scale "environments" of cells. This phenomenon was observed across various cell populations, including the red and white pulp and B cells in the mouse spleen, and all cells in the human lungs. When the relative differences in density are large, we turn to other vectorisation methods.

A stable vectorisation for persistence diagrams, persistence images, offers great flexibility in extracting information at various spatial scales [27,29]. We have shown how differences in large scale features observed in the PWDS are captured by persistence images weighted by persistence and birth values. However, we noticed how careful consideration is required when selecting weights in data sets with a large proportion of small-scale features. We observed that for certain point clouds, like the red pulp, the number of small-scale features (represented as light-red triangles in the PWDS) was so large that the weight $w_1(b,p) = p$ failed to emphasize the large-scale features adequately (see S4 Fig). In these cases, the results were similar to those obtained with a uniform weight $w_{\text{flat}}$. A much stronger weight, such as $w_2(b,p) = p^2$, was necessary for the large scale features to have a significant signal in the image. A similar effect happened with $w_3(b,p) = bp$ and $w_4(b,p) = bp^2$.

The third class of vectorisations we considered are obtained from elementary statistics. They are computationally efficient and low-dimensional, and they provided separation of the healthy and diseased groups in both tissue data sets. Unfortunately, they are less interpretable than Betti curves and persistence images. Moreover, they are challenging to visualise in a way that provides insight into the data.

For tissue with large cell numbers, topological analysis requires building computationally tractable higher-dimensional graphs (complexes). We compare witness complexes and alpha complexes and we find no clear advantage for the former. The computation of witness complexes was more time-consuming and it did not improve upon the results obtained using alpha complexes.

Through a suite of topological vectorisations and visualisations, we found that cell localisation patterns across multiple spatial scales are relevant for disease progression. In lupus spleen, the quantitative topological findings align with previous measurements of the dissipation of the marginal zone (which separates red and white pulp), the disintegration of the PALS (part of the white pulp), and the infiltration of red pulp by erythroblasts [20,30], providing a visualisation (Fig 5) and quantification of the resulting rearrangement of red and white pulp cells (Figs 6A-H and S4). Betti curves captured the progressive disorganisation of B follicles (Fig 6I-L), which aligns with and expands previous findings based on ring structures [20], as well as changes in the spatial distribution of other cell types (S3 Fig) including the clustering of megakaryocytes in lupus, a pattern not reported in [20].

For COVID-19 lung tissue, topological descriptors captured cell infiltration into alveolar spaces (Fig 7), consistent with established lung pathology [31,32], and revealed that spatial rearrangements of Endothelial cells correlate with disease staging (Fig 8), a pattern not explored in prior co-localisation studies [21]. By quantifying these structural changes, we move beyond qualitative or local assessments to provide global, interpretable descriptors that refine and extend traditional pathology, underscoring the potential of topological data analysis to reveal mechanisms of disease progression and support spatial biology.

## Supporting information

**S1 Fig. COVID-19 human lungs cell centroid data.** A: Spatial distribution of cell centroids for the 25 most common cell types (out of 50) in the COVID-19 human lung IMC data set, shown for six representative samples out of 32 total. The data set includes 32 lung sections, categorised by disease stage. We display one of the four healthy samples, two of the ten in the alveolitis (ALV) stage, two of the ten in the diffuse alveolar damage (DAD) stage, and one of the eight in the organising pneumonia (OP) stage. B: Cell counts of the two broader types of

cells: structural and immune, for the 32 samples in the data set, together with the spatial distribution of the two types of cells in the six representative samples from above. Note that the sizes of the samples vary.
(TIF)

**S2 Fig. Witness filtration.** Example of the witness filtration for a point cloud forming a loop. Landmarks points, in black, are chosen randomly. The persistence diagram contains only one point in each degree, corresponding to a single connected component and the loop, both formed at $\epsilon = 0$.
(TIF)

**S1 Table. Proportion of randomly chosen landmark points in witness filtrations for each point cloud.**
(XLSX)

**S2 Table. Parameter values for the persistence images.**
(XLSX)

**S3 Fig. Betti curves of CD4(+) T cells, B220(+) DN T cells and megakaryocytes populations in the murine spleen tissue.** A: CD4(+) T cells locations in the lupus murine spleen CODEX data. B: CD4(+) T cells normalised Betti-1 curves. The differences in the location of the peak indicate an decrease of density within the dense regions of the CD4(+) T cells in disease. One of the early diseased samples shows a behaviour different from the rest, due to the absence of dense regions. C: Cell counts of the CD4(+) T cells for each sample. For this cell type, the total cell counts do not change significantly between health and disease (Welch t-test p-value is 0.16). D: PCA plot of the CD4(+) T cells normalised Betti curves of degree 1 of the alpha filtration. E, F, G, H: same plots for B220(+) DN T cells. In this case the change in the location of peaks happens gradually as the disease progresses, due to the appearance and growth of large dense regions. We observe multiple peaks in the early diseased curves. I, J, K, L: same plots for megakaryocytes, but for the case of Betti-0 curves, which count the proportion of clusters which are alive at a given parameter value. We observe a very sharp decrease in the disease curves at small values, which is due to the presence of clusters in the cell population cells.
(TIF)

**S4 Fig. Persistence images with five different weightings for the red pulp in the murine spleen tissue.** Examples of persistence images with five different weighting functions for the red pulp with alpha filtration (BALBc-1, MRL-4). Here we observe how weights that depend on persistence, $p$, and/or birth value, $b$, "see" the large-scale features in the diagram. For the case of red pulp, the presence of a large number of small-scale features causes $w_1$ to yield very similar images to $w_{\text{flat}}$. A more intense weighting on persistence, like $w_3$ or $w_4$, is needed in order to highlight the large-scale. We include the PCAs of the persistence images for each weighting function.
(TIF)

**S5 Fig. Distribution of degree 1 persistence values for CD11c(+) B cells.** A: CD11c(+) B cells locations in the lupus murine spleen CODEX data. We see this cell type forms denser regions in the healthy samples, and is more sparsely distributed in the diseased ones. This is detected by three topological descriptors of degree 0, which yield correct 2-clusterings (see last rows of S3 Table). However, there is no visually clear distinction between early and intermediate samples. B: Cell counts of the CD11c(+) B cells for each sample. C: To further study the distribution of persistence values, we plot the cumulative distribution function of degree 1

persistence values for CD11c(+) B cells. Visually, these curves do not form 4 distinct groups, corresponding to the 4 stages. We note, however, that on the 90th percentile (when curves hit value 0.9) the three early stage curves are much closer and separated from the two intermediate stage curves. D: PCA plot of the CD11c(+) B cells topological descriptor obtained from the 10th, 25th, 50th, 75th and 90th percentiles of degree 1 persistence values, whose 4-clustering coincides with the stages of the disease. We observe that clusters are separable by their PC1 coordinates, and we check that the PC1 component is dominated by the 90th percentile (with a weight of 0.96). Hence, the coincidence of the early diseased curves on the 90th percentile we observed in B explains the clustering result. E: PCA plot of the CD11c(+) B cells topological descriptor obtained from cumulative distribution functions in B. To show that the cumulative distribution functions do not in fact separate in 4 groups, we use 4-means clustering directly on the curves. We obtain clusters that separate healthy and late stages correctly from the rest, but mix early and intermediate curves. We conclude that the correct 4-clustering with percentiles was an artifact of the choice of high percentiles, which are unstable for small populations.
(TIF)

**S3 Table. Topological descriptors obtained from the lupus murine spleen data set which yield a correct 2- or 4-clustering into disease stages, sorted by density of the cell type, degree of homology and silhouette score of the 2-clustering.** In the first column, we include the average number of cells in that type of point cloud across all samples. We also include the average number of landmarks, if the witness complex is computed for that cell type.
(XLSX)

**S4 Table. Topological descriptors obtained from the COVID-19 human lungs data set which yield a correct 2-clustering into healthy and diseased samples, or whose 4-clustering separates healthy samples from the rest.** In the first case we include the 2-clustering silhouette score in the last column. Next to each cell type we include the average number of cells of that type across samples, as well as the average number of landmarks, if the witness complex is computed for that cell type.
(XLSX)

**S5 Table. Topological descriptors obtained from the COVID-19 human lungs data set with the best Rand score compared to the clinical classification into disease stages.** Next to each cell type we include the average number of cells of that type across samples, as well as the average number of landmarks, if the witness complex is computed for that cell type.
(XLSX)

## Author contributions

**Conceptualization:** Maria Torras-Pérez, Iris H. R. Yoon, Helen M. Byrne, Ulrike Tillmann, Heather A. Harrington.

**Data curation:** Maria Torras-Pérez, Iris H. R. Yoon, Praveen Weeratunga, Ling-Pei Ho.

**Formal analysis:** Maria Torras-Pérez, Iris H. R. Yoon.

**Funding acquisition:** Maria Torras-Pérez, Helen M. Byrne, Ulrike Tillmann, Heather A. Harrington.

**Investigation:** Maria Torras-Pérez, Iris H. R. Yoon, Praveen Weeratunga, Ling-Pei Ho, Helen M. Byrne, Ulrike Tillmann, Heather A. Harrington.

**Methodology:** Maria Torras-Pérez, Iris H. R. Yoon, Ulrike Tillmann, Heather A. Harrington.

**Software:** Iris H. R. Yoon.

**Supervision:** Ling-Pei Ho, Helen M. Byrne, Ulrike Tillmann, Heather A. Harrington.

**Validation:** Maria Torras-Pérez, Ling-Pei Ho.

**Visualization:** Maria Torras-Pérez.

**Writing – original draft:** Maria Torras-Pérez, Iris H. R. Yoon, Ulrike Tillmann, Heather A. Harrington.

**Writing – review & editing:** Maria Torras-Pérez, Iris H. R. Yoon, Praveen Weeratunga, Ling-Pei Ho, Helen M. Byrne, Ulrike Tillmann, Heather A. Harrington.

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
