## [Decision Letter · Decision Letter 0]

6 Jul 2025

PCOMPBIOL-D-25-00887

Topology across scales on heterogeneous cell data

PLOS Computational Biology

Dear Dr. Harrington,

Thank you very much for submitting your manuscript to PLOS Computational Biology. After careful consideration, we feel that it has merit to be published in our journal but as it stands currently it does not fully meet PLOS Computational Biology's publication criteria. Therefore, we kindly invite you to submit a revised version of the manuscript that addresses the points raised during the review process.

Please submit your revised manuscript within 30 days Sep 05 2025 11:59PM. If you will need more time than this to complete your revisions, please reply to this message or contact the journal office at ploscompbiol@plos.org. Please include the following items when submitting your revised manuscript:

We look forward to receiving your revised manuscript.

Kind regards,

Calina Copos, Ph.D.

Academic Editor

PLOS Computational Biology

Stacey Finley, Ph.D.

Section Editor

PLOS Computational Biology

**Additional Editor Comments:**

The reviewers raise suggestions to better clarify parts of the manuscript and more clearly describe the biological implications of this work. These points should be addressed in the revised submission.

**Journal Requirements:**

**Reviewers' comments:**

Reviewer's Responses to Questions

**Comments to the Authors:**

Reviewer #1: Review of “Topology across scales on heterogeneous cell data”

This is an exemplary methods paper that details two case studies in applying techniques of topological data analysis (TDA) to multiplexed images of biological tissues. In particular, it addresses the challenge of quantifying spatial distributions of different cell types within a single tissue sample and whether this information correctly classifies a tissue as healthy or diseased. The authors identify particular cell-types whose spatial distribution is associated with healthy versus diseased states in lupus murine spleen and human COVID-19 infected lung.

The main contribution of this work is in the development of practical, reproducible workflows applying TDA to real bio-medical images. I particularly appreciate the use of standard TDA tools that are widely accessible and not overly complicated. This should enhance the wider relevance and uptake of the methods, which are relevant to any 2D spatial point-clouds with different point types.

The authors have made a thorough and comprehensive test of different methods to vectorise persistent homology data. There was no clear “winner” in the 2-label clustering tasks. The authors make some general conclusions about when different vectorisations are more or less suitable, and argue that weighted persistence image vectorisations provide a flexible method when comparing samples with large differences in point density.

The authors introduce two new methods to the TDA toolbox. The first is an effective visualisation tool that plots a 2-simplex (triangle) when it is one that causes the “death” of a hole, with a color and intensity determined by birth-value and persistence lifetime. The second is the introduction of a weighting function to the persistence image vectorisation step, that allows researchers to “tune” the relative importance of different parts of the persistence diagram.

The data is provided in a github repository. Quantitative analysis tools are based on open-source software packages and clearly referenced. The workflow is clearly described and should be reproducible.

Additional points to address:

(a) I would like the authors to include one more data field in Tables 2,3,4: could they please add a column with the number of points in each point cloud, and the number of witness points used when building the witness filtrations.

(b) On page 16, I would like the authors to clarify what they mean by “correct clustering”? Does this mean all labelled data are correctly allocated to a respective cluster with respect to a given TDA vectorisation? (this seems too strong). Or is it that the silhouette score is positive? (this seems rather generous).

(c) Also in that section, there is no discussion of the second cell type that gave a correct 4-clustering of the lupus samples (CD11c(+) B cells) using a statistical percentile of persistence vectorisation. The score for this 4-clustering is 0.4996, which is stronger than that for the plain B cells with Betti-1 curve vectorisation with 0.3319. Could the authors comment on this?

Minor edits:

1. A sentence on page 16 seems incomplete/ not properly integrated with the text: “Two cell types that give a significant spatial pattern between disease stages.”

2. The abstract has “persistence homology” but the standard term is “persistent homology”.

3. Check the journal style preference regarding the convention that numbers smaller than ten are written as words and those larger as digits?

eg I would rewrite “We next analyse all 25 cell types and 2 broader groups of cell types” as

“We next analyse all 25 cell types and two broader groups of cell types”,

but later in that paragraph keep “2-clustering” as is.

Reviewer #2: The paper concerns development of new tools to analyse multiplex spatial imaging data derived from biological experiments. In particular it uses notions from topological data analysis (TDA) to help incorporate information at different spatial scales. The paper is very well written and on a topic of current interest and importance.

One of their main innovations is the introduction of a new idea, termed persistence weighted death simplices (PWDS), that allows them to visualise and interpret data features. To demonstrate this idea they use two previously published spatial datasets, which come with annotations. One of the data sets is summarised in Figure 2 which helps orient the reader but I could not see a similar summary of the other data. Perhaps that is something to include for completeness.

The paper includes a nice outline of persistent homology and the various tools that they use. This is mostly not new material but it may be unfamiliar to many readers and it is well explained so I think that it is appropriate to include it here. This section leads into a description of their main technical innovation, the PWDS. This is intuitively a nice idea and it produces images that are readily interpretable — much more immediately than the persistence diagrams, which require some expert interpretation. This idea is a strong part of the paper in my opinion. They have also chosen a good colour palette that will be easily distinguishable to most users, including those that are colour-blind.

Their main results, which consist of detailed TDA analysis of the test data sets, are clear and rigorous. The main conclusions are quite technical however, and my main query concerns the additional insight that their analysis provides.

Do the topological features they identify only serve to highlight characteristics of the diseases under study that were already known (i.e. are the results a mathematical way of saying what a pathologist would already know)?

If yes then do they shed any extra light on those characteristics (could they provide quantitative metrics that might help practitioners interpret pathology images, for example)?

If no (i.e. their analysis uncovers biological characteristics that were unknown), then are those characteristics amenable to further exploration?

I think that while technically excellent the paper would be improved with some more discussion of the biological meaning of their analysis and how it relates to what is already known about the diseases under study.

Overall I think that this is a very strong study that will make a good addition to the literature in this area.

**Have the authors made all data and (if applicable) computational code underlying the findings in their manuscript fully available?**

Reviewer #1: Yes

Reviewer #2: Yes

PLOS authors have the option to publish the peer review history of their article (what does this mean?). If published, this will include your full peer review and any attached files.

Reviewer #1: No

Reviewer #2: No

**Figure resubmission:**
---

## [Editor Report · Decision Letter 1]

24 Aug 2025

Dear Dr Harrington,

Thank you for addressing all of the reviewers' comments. We are pleased to inform you that your manuscript 'Topology across scales on heterogeneous cell data' has been provisionally accepted for publication in PLOS Computational Biology. 

Best regards,

Calina Copos, Ph.D.

Academic Editor

PLOS Computational Biology

Stacey Finley, Ph.D.

Section Editor

PLOS Computational Biology

---

## [Editor Report · Acceptance letter]

PCOMPBIOL-D-25-00887R1

Topology across scales on heterogeneous cell data

Dear Dr Harrington,

I am pleased to inform you that your manuscript has been formally accepted for publication in PLOS Computational Biology. Your manuscript is now with our production department and you will be notified of the publication date in due course.

With kind regards,

Anita Estes
